# Usability Evaluation of Slanted Computer Mice

**DOI:** 10.3390/ijerph18083854

**Published:** 2021-04-07

**Authors:** Miguel L. Lourenço, Fátima Lanhoso, Denis A. Coelho

**Affiliations:** 1Research Unit for Inland Development, Guarda Polytechnic Institute, 6300-559 Guarda, Portugal; 2Centre for Mechanical and Aerospace Science and Technologies, Department of Electromechanical Engineering, Universidade da Beira Interior, 6201-001 Covilhã, Portugal; fatimalanhoso1964@gmail.com; 3Human Factors and Ergonomics Research Group, Department of Supply Chain and Operations Management, School of Engineering, Jönköping University, 551-11 Jönköping, Sweden

**Keywords:** satisfaction, ease of use, computer mouse, preference, effort, discomfort, efficiency, effectiveness, usability, slanted pointing device

## Abstract

Prevention of musculoskeletal disorders is supported by use of slanted rather than horizontal pointing devices, but user acceptance of the former may be compromised due to lower perceived ease of use. This study compares subjectively rated usability (N = 37) for three sizes of slanted computer mice and includes a horizontal small conventional device as a reference. For a random subset of the sample (*n* = 10), objective usability parameters were also elicited. Participants followed a standard protocol which is based on executing graphical pointing, steering, and dragging tasks generated by a purpose-built software. Subjective ratings were collected for each of the four pointing devices tested. The three slanted devices differed in size but were chosen because of an approximately similar slant angle (around 50–60 degrees relative to the horizontal plane). Additionally, effectiveness and efficiency were objectively calculated based on data recorded for the graphical tasks’ software for a random subset of the participants (*n* = 10). The results unveil small differences in preference in some of the subjective usability parameters across hand size groups. This notwithstanding, the objective efficiency results are aligned with the subjective results, indicating consistency with the hypothesis that smaller slanted devices relative to the user’s hand size are easier to use than larger ones. Mean values of weighted efficiency recorded in the study range from 68% to 75%, with differences across devices coherent with preference rank orders.

## 1. Introduction

An enduring conflict between favorable long-term health outcomes and short-term usability results persists around the use of computer mice. Pronation eliminating, vertical pointing devices are recommended for the prevention of musculoskeletal disorders of the forearm and wrist [1], but their usability may be compromised by the position of the buttons hidden from view, which for many users may consist in a deterrent to acceptance of the alternative shaped computer handheld pointing devices in the short term. Previous research identified usability advantages for slanted devices, compared to vertical pointing devices but also indicated that the relative size of the device in relation to the hand size is a factor influencing measured objective usability [2,3]. Previous studies have also suggested that smaller devices relative to the size of the hand promoted more efficiency in use than larger devices [4,5]. Moreover, more dynamic activity of the muscles activated in the use of the computer ensues in smaller devices, relative to the size of the user’s hand, as has been registered for slanted, vertical, and horizontal devices [3].

The conventional (horizontal) PC mouse leads the user to uphold an unnatural posture, forcing the forearm to pronate until the end of the joint rotation amplitude. The PC mouse is most likely the most widely used hand tool in the world; the conventional geometry appeared about 50 years ago and the discomfort verified during its use was considered normal, despite the forearm and wrist posture that had to be adopted contrasting with the handwriting posture, learned at school from an early age. A slanted PC mouse, when properly sized, leads to a more neutral posture of the forearm, between pronation and supination. “Good design enables equipment to be used with the joints in the middle of their range of movement” [6]. Moreover, concepts such as ease of use and satisfaction, in addition to effectiveness and efficiency, are considered an integral part of a broader concept called usability [7]. A slanted handheld pointing device, provided it is properly dimensioned, will promote a more neutral posture of the forearm and wrist, with less effort required to operate and less discomfort being felt in its use compared to the conventional PC mouse, providing a usability advantage. In alignment with the principles of ergonomics, this study seeks to contribute broadly to improving working conditions regarding human well-being, safety, and health [8]. The degree of inclination is also an important usability factor in this type of handheld device [2,9,10].

As for applicable guidelines regarding usability, the ISO 9241-11 (2018) standard: Ergonomics of human-system interaction [7] presents concepts and definitions in this context. This standard defines usability as “the extent to which a system, product or service can be used by specified users to achieve specified goals with effectiveness, efficiency and satisfaction in a specified context of use”, providing annexes A and B that inform about how usability can be considered for different scopes of contexts of use and offer examples of usability measures. On the other hand, the ISO 9241-400 (2007) standard: Ergonomics of human-system interaction [11] refers to the principles and requirements for physical input devices. However, it is considered that it does not have a sufficiently structured process, which facilitates its application, to evaluate the usability of computer mice. Thus, regarding assessing the usability of this type of devices, the authors developed specific tools to assess effectiveness, efficiency, and satisfaction [2,5,10].

The current study builds on the previous research developed and aims comparing the effect of differing sizes of slanted computer mice on subjectively rated usability and includes a horizontal small device as a reference in the comparison. The underlying hypothesis for this study is that smaller slanted devices relative to the user’s hand size are easier to use than larger ones. An indication of the criteria for optimizing an individual fit with handheld pointing device, that is simultaneously easier to use, would translate into greater acceptability and better long-term health outcomes for people involved in activities where intensive use of handheld pointing devices ensues. One of the most pervasive examples of such an activity is office work with visual display terminals [6]; another one is the computer aided design activity [12].

Conceptually, and from a theoretical perspective, usability concerns the rich interaction of users and technology in a socio-technical context [13]. Usability may even be viewed as part of the higher-level concept of interaction experience, together with user experience (including aesthetics, trust, value, affect) and accessibility [14]. Bringing the scope to the application at hand, computer handheld pointing devices, within this realm, it is widely accepted to consider, as part of the usability concept, the aspects of satisfaction as well as performance, which is viewed as composed of both effectiveness and efficiency [11,15]. In this study, effectiveness and efficiency were calculated, and discomfort, effort, performance, and ease of use were subjectively assessed.

## 2. Materials and Methods

### 2.1. Study Design

The order of user testing of the four computer mice was randomized for every subject. They followed a standard protocol which is based on executing graphical pointing, steering, and dragging tasks generated by a purpose-built software [16], as developed and described in previous publications [2,3,10]. Subjective ratings of satisfaction, performance, effort, ease of use, as well as forearm and hand discomfort were collected for each of the four pointing devices tested. Additionally, effectiveness and efficiency were objectively calculated based on data recorded on the graphical tasks software that were processed for a random subset of the participants (*n* = 10; 2 females and 8 males). The three slanted devices differed in size but were chosen because of an approximately similar slant angle (around 50–60 degrees relative to the horizontal plane). The devices are represented in images embedded in Table 1. Participants were instructed to only use the left and right main buttons as well as the middle button which were similar across all the devices, even if some had additional buttons that did not activate the graphical tests, however.

### 2.2. Variables

The purpose-built software [16] collected several parameters of the trials including time to complete tasks and errors undergone, enabling calculation of effectiveness and efficiency usability parameters. The effectiveness (*efa*) for pointing and dragging tasks was calculated from Equation (1) whereas for the steering task Equation (2) was used. Efficiency (*efi*) was calculated from Equation (3). The equations spring from previous work of the lead authors [2,5,10]. These equations are based on error rate (failures in hitting targets divided by total number of targets in the task) and time to complete tasks. Note that in equation (1), the number of failed targets is always less or equal than the total number of targets. The steering task represents an exception, in that there are no targets identified, but the deviation from the shortest path is considered, akin to a grade of relative failure:(1)$$efa_{\left(point\;\wedge \;drag\right)}=1-\frac{No.FailedTargets}{No.TotalTargets},$$
(2)$$efa_{\left(steering\right)}=\frac{minimum\;mean\;deviation}{mean\;deviation\left(subject\right)},$$
(3)$$efi_{\left(point\;\wedge \;drag\;\wedge \;steering\right)}=efa\times \frac{minimum\;mean\;completion\;TIME}{mean\;completion\;TIME\;\left(subject\right)},$$

*efa_(point ˄ drag)_*—effectiveness of pointing and effectiveness of dragging,

*efa_(steering)_*—effectiveness of steering,

*efi_(point ˄ drag ˄ steering)_*—efficiency of pointing (dragging and steering),

*No. FailedTargets*—number of failed targets by the subject for the particular task,

*No. TotalTargets*—total number of targets to be hit for the particular task,

*minimum mean deviation*—lowest mean deviation across the whole set of replications of subject-device combinations,

*mean deviation(subject)*—mean deviation achieved in the steering task for the subject-device combination,

*minimum mean completion TIME*—lowest mean completion time across the whole set of replications of participant-device combinations (for the particular task),

*mean completion TIME (subject)*—mean time to complete the particular task for the participant-device combination.

An objective measure for internal validation of the self-assessed dimensions of usability presented in the previous section, considering the overarching aim of fostering greater acceptance of health promoting alternative devices, by fine-tuning recommendations for the public at large, in line with the chances of fostering greater acceptance for the slanted computer handheld pointing devices was sought. Based on previous research [2,3], an efficiency index was applied to compound the efficiency objective measures across the six graphical tax into one indicator, as shown in Equation (4). The development of such efficiency index was based on computer aided design operators and yielded relative percentage of time for the tasks in the order of 10 to 20%. The pre-existing coefficients were retrieved from earlier experimental studies [2,3,10] and are shown in Table 2:(4)$$efi_{w}=\frac{a.efi_{poi_{l}}+b.efi_{poi_{m}}+c.efi_{poi_{s}}+d.efi_{drag_{l}}+e.efi_{drag_{m}}+f.efi_{st}}{a+b+c+d+e+f},$$

$efi_{w}$—weighted efficiency,

$efi_{poi_{l}}$—efficiency of pointing large,

$efi_{poi_{m}}$—efficiency of pointing medium,

$efi_{poi_{s}}$—efficiency of pointing small,

$efi_{drag_{l}}$—efficiency of dragging left,

$efi_{drag_{m}}$—efficiency of dragging middle,

$efi_{st}$—efficiency of steering,

*a*, *b*, *c*, *d*, *e*, and *f*—fractional time of use of each type of operation.

### 2.3. Participants

Participants (N = 37; 15 females and 22 males) were recruited from right-handed undergraduate students taking courses in Human Factors and Ergonomics for course credit. All had normal or corrected to normal vision. Participant mean age was 23.1 years, with a standard deviation of 6.5 years. The set of participants included Portuguese nationals as well as nationals from Portuguese speaking countries studying at the University of Beira Interior in Portugal at the time of the study’s data collection phase (May 2018).

### 2.4. Test Tasks

The tasks of pointing and selecting graphic entities, presented by Odell and Johnson [17], satisfy the requirements resulting from mouse operations carried out in office activity. In that study, 12 participants tested various computer mice performing pointing tasks. These tasks present three levels of difficulty according to the dimensions of the circular targets, which are larger in the task of pointing large and decrease in size in the other two tasks (medium and small). In each task, one of 18 circular targets is activated randomly and only ‘turns off’ when selected by the mouse cursor, after which the diametrically opposite circle (at 180°) is activated, which then only turns off when selected, and then another random circle is activated. The dimensions of the targets in the pointing large task are like those of the folders and icons on a desktop computer, and the size of the targets in the pointing small task is like the size of smaller characters.

In the work reported by Houwink et al. [18], 30 participants tested two computer mice while performing pointing, dragging, and steering tasks. Each omnidirectional task took about a minute to complete. In these tasks, the distribution of targets in space took on a geometry like the one used by Odell and Johnson [17], despite assuming much larger distances between diametrically opposing centers. In the dragging task, subjects had to select the active target (by clicking the mouse button) and drag it to the diametrically opposite circle by releasing the mouse button on the target. In the steering task, subjects had to move, as rectilinearly as possible, the active target from one point to another within the boundaries of a two-dimensional ‘tunnel’.

Lee et al. [19] tested alternative ways to click the left mouse button (adapted model), using pointing, dragging, and steering tasks. These authors considered 15 circular targets of 7.5 mm diameter for the pointing task, a sequence of 20 horizontal tunnels arranged horizontally and vertically with different lengths and, or widths for the steering task and selecting and dragging shapes by fitting them into their ‘shadows’ in the dragging task.

To test the devices of interest, we chose to structure the test tasks in the following way: pointing large, medium, and small tasks configured identically to Odell and Johnson [17] and dragging and steering modes configured appropriately from the configuration reported by Houwink et al. [18]. The dragging test tasks sequentially use the left, middle, and the right mouse buttons. Figure 1 illustrates all the tasks. In the tasks of pointing, the test is initiated by the participant when she or he clicks on the center circle and the action takes place as in Odell and Johnson [17], except for the difficulty increasing in sequence, starting in the pointing large and ending in the pointing small task. The steering task had targets of 7 mm diameter, like the dragging task, and the goal of maintaining the rectilinear trajectory of the displacement.

Instructions given to the participants were to complete the tasks as well as they could, in the shortest amount of time. The tasks were designed with an approximate duration of one minute. However, the number of mistakes made by the participants influences the duration of the respective task, due to the requirement for its completion (number of targets).

### 2.5. Data Analysis

Effectiveness and efficiency were objectively calculated for a random subset of the participants (*n* = 10; 2 females and 8 males) and compared to the subjective results presented, using non-parametric statistics [20] supported by IBM SPSS v.23. (IBM, New York, NY, USA), Kendall’s W was used to assess the level of concordance between rank order preferences indicated by participants for a number of subjective dimensions. The *p*-value for all statistical tests was set at 0.05. Moreover, Pearson correlation factor was applied as a measure of association across pairs of variables, including for pairing subjective and objective measures collected in the study.

## 3. Results

Participant hand anthropometric parameter information is shown in Table 3 and includes an auxiliary hand size parameter [2,3,5,10]. Hand size and sex show a Pearson correlation factor of 0.831 (*p* < 0.001); most subjects with hand size below the average (*n* = 14) for the group were female, and most subjects with hand size above the average (*n* = 23) were male.

Subjective outcomes were obtained for the entire sample of participants (N = 37) immediately after performing the standardized graphical tests. Both preference order rankings and absolute ratings are reported in the first subsection. For a randomly selected subset of the participants (*n* = 10), objective outcomes are reported in the second subsection. These were calculated based on the parameters recorded in real-time by the custom-made graphical task generation software [16] and according to previously reported procedures [2,10]. Based on this latter subset of participants, association is established between the subjective and objective outcomes and is reported in the third subsection.

### 3.1. Preference Order Rankings and Subjective Ratings

For the overall sample (N = 37), significant concordance was attained for the preference order ranking of the following subjective dimensions:Effectiveness preference ranking order was small horizontal first, medium slanted second, small slanted third, and large slanted fourth (Kendall’s W = 0.178; *p* < 0.001).Effort preference ranking order: small horizontal, small slanted, medium slanted, large slanted (Kendall’s W = 0.164; *p* < 0.001).Performance preference ranking order: small horizontal, small slanted, medium slanted, large slanted (Kendall’s W = 0.157; *p* = 0.001).Aesthetic preference ranking order: medium slanted, large slanted, small slanted, small horizontal (Kendall’s W = 0.178; *p* < 0.001).Ease of use preference ranking order: small horizontal, medium slanted, small slanted, large slanted (Kendall’s W = 0.148; *p* = 0.001).Satisfaction preference ranking order: small horizontal, small slanted, medium slanted, large slanted (Kendall’s W = 0.073; *p* = 0.045).

The overall preference ranking regarding discomfort among the devices approached significance and consists of: small slanted first (less discomfort), small horizontal second, medium slanted third and large slanted fourth (most discomfort) (Kendall’s W= 0.062; *p* = 0.076). Comparative results of mean ranking of preference for small hand size versus large hand size groups are shown in Table 4. The results unveiled differences in preference in some of the subjective usability parameters across hand size groups. This notwithstanding, the overall picture of the results across hand size groups is quite similar, with small differences in preference across hand size groups. While the subjective rankings of preference encompass user experience, as well as usability aspects, the satisfaction ranking achieves a significant level of concordance for the smaller hand size group, placing the small, slanted device before the horizontal device. The preference ranking order for less uncomfortable device, follows suit for this hand size group as well. Only three out of the 14 rank orders elicited does not show concordance among the participants, and all three concern the larger hand size group (less discomfort preference, less effort preference, and satisfaction preference).

Ratings of satisfaction, performance, ease of use, effort, hand discomfort, and forearm discomfort are shown graphically in Figure 2, clustered by hand size groups and device. Pearson correlation factors across the variables depicted in Figure 2 are shown in Table 5 (*p* < 0.001). Differences across hand size groups are more striking for the smaller devices, especially for the dimensions of satisfaction, performance, and effortlessness. In the dimensions of ease of use, and lack of hand discomfort, the clustered bars in Figure 2 display higher ratings on average given by the large hand size group to the medium and large slanted devices, but with higher ratings on average given by the smaller hand size group to the small slanted and small horizontal devices. Lack of forearm discomfort is consistently rated higher for the large hand group than for the small hand group on average for all the slanted devices. These results altogether suggest a perceived better fit across most subjective dimensions of the smaller devices to the smaller hand size group, and the medium and large devices to the larger hand size group.

Correlation analysis among the dimensions subjectively rated suggests very high consistency between subjective self-assessed performance and satisfaction, as well as effortlessness. The lowest level of association is found between lack of forearm discomfort and ease of use, followed by lack of forearm discomfort and satisfaction, albeit with about 50% of co-variance among both pairs. Co-variance for the remaining pairs is in the range of 60 to 70%. These subjective experimental results indicate more co-variance of lack of hand discomfort than lack of forearm discomfort with ease of use, performance, and satisfaction. Conversely, lack of forearm discomfort shows slightly more co-variance with effortlessness than does lack of hand discomfort.

### 3.2. Objective Measures

The mean and standard deviation values for weighted efficiency results computed for a random subset of 10 participants (8 males) are shown per device in Table 6. Standard deviation is only calculated for the 10 participants altogether, as well as for male participants in the subset, but not for female due to reduced sample size.

Overall, results indicate larger weighted efficiency for the small horizontal device, followed by the small slanted, the large slanted, and the medium slanted device. Across the board, large hand size participants score higher mean values of weighted efficiency than smaller hand size participants, with the only exception of the medium slanted device. The mean results for the eight large hand size participants in the subset are consistent with the underlying assumption that smaller devices relative to the size of the hand promoted more efficiency in use than larger devices.

### 3.3. Association between Subjective and Objective Outcomes

The subset of results for the ten participants for which objective efficiency results were obtained as shown in the previous subsection were correlated with the subjective ratings for the same subset. Statistical significance was set at *p*-value below 0.05. Pearson correlation with statistical significance was found for the pair efficiency of dragging with the left button and satisfaction (r = 0.327, *p* = 0.039). Moreover, the association between the variable pair efficiency of pointing large and subjectively rated performance approached significance as well (r = 0.309, *p* = 0.052).

Restricting to the subset with large hand size yields, more expressive association between the objective and subjective evaluation measures collected in the data, even if the number of participants involved diminishes to only eight. The results of association achieving statistical significance are shown in Table 7 for this restricted universe of participants. From the objective measures set, it is efficiency of pointing large that correlates with more subjective usability evaluation measures. Conversely, effortlessness is the subjective measure with a broader set of statistically significant associations with objective measures. Despite the reduced number of subjects (*n* = 8), the associations unveiled with statistical significance have moderate intensity, indicating a co-variance in the order of 40% for the pairs of variables involved. Albeit, limited, this provides an indication of alignment between the objective and subjective variables used in the study.

## 4. Discussion

The results presented for subjectively assessed measures of usability as well as for objectively assessed efficiency indicate alignment with previous studies, suggesting that smaller devices relative to the size of the hand promoted more efficiency in use than larger devices. Moreover, moderate values of association between subjective and objective variables support the internal validity of the experimental results reported in this study. Any other study that has tested different sizes of devices with similar inclined geometry, that is, with angular amplitude between 50 and 60 degrees, is unknown to the authors at the time of writing. Nevertheless, the results obtained here allow to establish a high level of qualitative similarity with the results obtained in references [5] and [10]. Both suggest that smaller PC mice compared to the users’ hand size allows reaching higher levels of usability.

Differentiating across the two categories of hand size, it can be seen for most indicators collected that the larger hand size group show more efficiency and preference for the medium slanted device, while the smaller hand size group gather around the smaller slanted device. Hence, there is broad alignment with the hypothesis that smaller slanted devices relative to the user’s hand size are easier to use than larger ones. This notwithstanding, there is a proportion effect suggesting that the one size fits all approach does not apply, and there is still a need for scaling and differing sizes of slanted computer handheld pointing devices to fit the user’s hand anthropometrics.

This study is not free of limitations; in particular, there is an imbalance between the number of participants in the hand size categories for which an objective assessment of usability in interaction with the four computer mice tested was done. This notwithstanding, the number of participants in the larger hand size subgroup with objective assessments was big enough for allowing some significant statistical associations to be elicited. Another limitation springs from the fact that the usability efficiency weighted indicator used in the study to aggregate the efficiency measures for the six tasks at hand, was developed taking into consideration computer aided design tasks and as such its generalizability to a broader set of office tasks is also limited.

Future research might include refining the efficiency weightings for specific activities from observational data, as well as performing the comparative assessments on a broader set of computer handheld slanted pointed devices. Moreover, it is envisaged that studying the muscular activation in the forearm, arm, and shoulder will provide further empirical evidence to support the development of recommendations aimed at improving the work environment and fostering well-being and health outcomes in the long run for users of pointing devices.

## 5. Conclusions

Slanted computer handheld pointing devices represent a compromise between the ease of use and wide acceptance and dissemination of the horizontal devices and the biomechanically and physiologically health promoting vertical devices, but with short-term reduced usability and user acceptance. To support developing occupational health recommendations for this kind of pointing devices, the current study performed empirically supported usability evaluation of three slanted devices, differing in size. The results support the acceptance of the hypothesis that smaller devices relative to the hand size are easier to use. Moreover, the subjective assessments also indicate a preference for smaller slanted devices that only rivals the acceptance of the conventional horizontal device, the widely disseminated paradigm by default, which is however mediated by hand size. This is indicated in the fine differences found between the larger and smaller hand size participants, with the former preferring medium-sized and the latter preferring smaller-sized slanted devices.

Salient quantitative results from the study include the high correlation factor (0.822) verified among subjective performance and satisfaction usability parameters. This is contrasted by smaller values of correlation factor when associating objective and subjective usability parameters (e.g., weighted efficiency versus effortlessness with r = 0.418 and a *p*-value of 0.017). Mean values of weighted efficiency recorded in the study range from 68% to 75%, showing small differences across devices but which are inherently coherent with the preference rank orders.

The study reported in this article was designed with the goal of identifying the effect of size on usability for slanted devices. Considering the results obtained in terms of subjectively evaluated usability dimensions, and the results obtained from calculated efficiency, the study suggests that for the same inclination it is preferable to choose a smaller size. This conclusion is nuanced by the proportionality rule, as it is the medium-sized slanted device that obtains the best results in the evaluations related to the larger hand size group and the small-sized slanted device that is better for the smaller hand size participants.

The following audiences may be interested in the results of this work: entities that develop and or that produce PC mice, users who make intensive use of PC mice, as well as regulators and work environment authorities establishing recommendations to promote working health and extended social sustainability.

## Figures and Tables

**Figure 1 ijerph-18-03854-f001:**
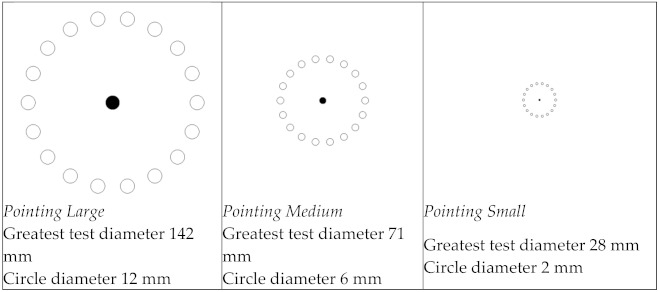
Test tasks (pointing, dragging, and steering).

**Figure 2 ijerph-18-03854-f002:**
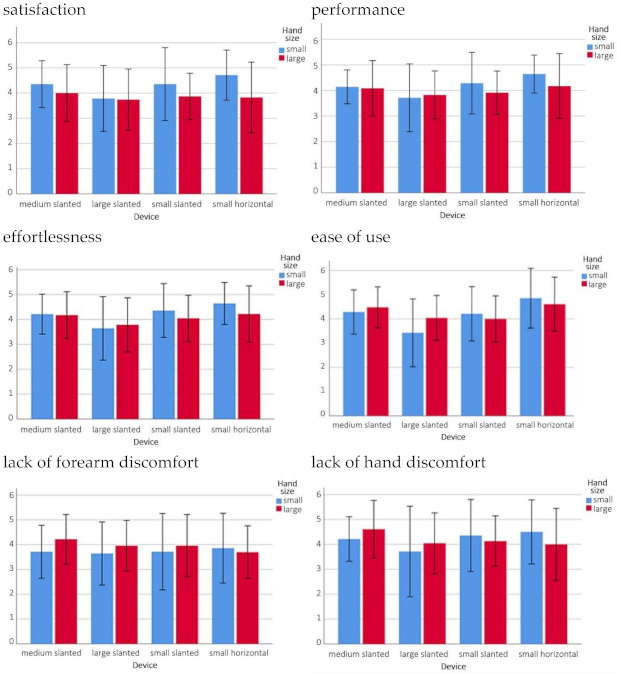
Clustered bar charts of the mean and standard deviation of subjectively rated satisfaction, efficiency, effort, ease of use, forearm discomfort, and hand discomfort by hand size and device [Higher values are more positive than lower values, e.g., satisfaction: 6–Very satisfied; 0–Not satisfied].

**Table 1 ijerph-18-03854-t001:** Overview of wireless handheld pointing devices (all right-handed) used in the study and under comparison.

Device.	Small Horizontal	Small Slanted	Medium Slanted	Large Slanted
Brand and model	Microsoft	Moko	CSL	Anker
Mobile 1850	S8	E.VE	98ANWVM

Picture	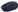	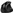	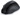	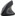
Dimensions (L × W × H) [mm]	100.0 × 58.1 × 38.2	104.9 × 72.9 × 59.9	126.0 × 68.1 × 62.0	120 × 62.8 × 74.8
Nr. of Buttons	3	6	6	5

**Table 2 ijerph-18-03854-t002:** Percentage values for fractional time of use coefficients [2,3,10].

Pointing Device Operations	Pointing at Large Targets	Pointing at Medium Targets	Pointing at Small Targets	Dragging with Left Button	Dragging with Middle Button	Steering
Fractional time of use coefficient	a	b	c	d	e	f
Percentage value	10.1%	21.8%	18.1%	23.1%	11.2%	15.7%

**Table 3 ijerph-18-03854-t003:** Hand anthropometry sample statistics (15 female, 22 male) (SD–standard deviation).

Dimension.	Hand Length [mm]		Hand Width [mm]		Hand Size (Width + Length)	
Sample	Mean	SD	Mean	SD	Mean	
Below hand size overall mean (*n* = 14; 13 female)	167.7	9.1	76.5	4.2	244.2	
Overall (N = 37)	179.6	11.7	82.5	82.5	262.1	
Above hand size overall mean (*n* = 23; 21 male)	186.9	5.4	86.2	3.3	273.1	

**Table 4 ijerph-18-03854-t004:** Small (*n* = 14) and Large (*n* = 23) hand size groups (below and above the hand size parameter mean for the entire sample) and preference ranking among devices for various subjective dimensions.

Dimension	Hand Size Group	1st	2nd	3rd	4th	Kendall’s W (*p*-Value)
Less discomfort preference	Small	Small slanted	Small horizontal	Medium slanted	Large slanted	0.188 (0.048)
	Large	Small horizontal	Small slanted	Large slanted	Medium slanted	0.025 (0.628)
Effectiveness preference	Small	Small horizontal	Tie between small and medium slanted		Large slanted	0.294 (0.006)
	Large	Small horizontal	Medium slanted	Small slanted	Large slanted	0.124 (0.035)
Performance preference	Small	Small horizontal	Small slanted	Medium slanted	Large slanted	0.239 (0.018)
	Large	Small horizontal	Small slanted	Medium slanted	Large slanted	0.143 (0.020)
Less effort preference	Small	Small slanted	Small horizontal	Large slanted	Medium slanted	0.173 (0.063)
	Large	Small horizontal	Small slanted	Medium slanted	Large slanted	0.025 (0.638)
Aesthetic preference	Small	Small slanted	Medium slanted	Large slanted	Small horizontal	0.367 (0.001)
	Large	Large slanted	Medium slanted	Small slanted	Small horizontal	0.186 (0.005)
Ease of use preference	Small	Small horizontal	Small slanted	Medium slanted	Large Slanted	0.198 (0.040)
	Large	Small horizontal	Medium slanted	Large Slanted	Small slanted	0.182 (0.006)
Satisfaction preference	Small	Small slanted	Small horizontal	Medium slanted	Large slanted	0.190 (0.047)
	Large	Small horizontal	Medium slanted	Small slanted	Large slanted	0.059 (0.257)

**Table 5 ijerph-18-03854-t005:** Pearson correlation factors (*p* < 0.001) for the subjective ratings (N = 37).

	Lack of Forearm Discomfort	Ease of Use	Effortlessness	Performance	Satisfaction
Lack of hand discomfort	0.571	0.647	0.638	0.693	0.691
Lack of forearm discomfort		0.492	0.668	0.591	0.555
Ease of use			0.661	0.717	0.719
Effortlessness				0.780	0.741
Performance					0.822

**Table 6 ijerph-18-03854-t006:** Mean weighted efficiency values obtained from purpose-built software [11] recorded objective data for a randomly selected subset of participants (*n* = 10); (standard deviation shown in parentheses where meaningful).

	Pointing Device	Medium Slanted	Large Slanted	Small Slanted	Small Horizontal
Sample Subset					
Selection (*n* = 10)		69.6% (6.9%)	71.3% (4.6%)	72.8% (4.2%)	73.6% (7.5%)
Large hand size (n_male_ = 8)		69.4% (6.6%)	71.8% (4.8%)	74.0% (3.7%)	75.0% (6.9%)
Small hand size (n_female_ = 2)		70.4% (-)	69.2% (-)	68.3% (-)	68.3% (-)

**Table 7 ijerph-18-03854-t007:** Pearson correlation factor values achieving significance (*p* < 0.05) for the association of objective and subjective usability evaluation measures, restricting to participants with large hand size (*n* = 8).

Variable	Ease of Use	Effortlessness	Performance
Weighted efficiency		r = 0.418 (*p* = 0.017)	
Efficiency of pointing large	r = 0.375 (*p* = 0.035)	r = 0.392 (*p* = 0.027)	r = 0.405 (*p* = 0.021)
Efficiency of steering		r = 0.395 (*p* = 0.025)

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
