# Peer review of "Usability Evaluation of Slanted Computer Mice"

_ijerph, 2021, doi:10.3390/ijerph18083854_

Round 1

Reviewer 1 Report

The article is very interesting and complete; however, it needs to be restructured.

In the materials and methods section, I suggest using a better structure: study design, variables, participants, test tasks, and data analysis. In lines 74-81, the authors mention some details about statistical procedures, but it is not clear what non-parametric test they used and the level of significance. In the results Kendall´s W is mentioned but not in the methods.

Try not to mix results in the materials and methods section. For example, Table 2 shows the results of hand anthropometry, and line 79 shows the result of the Pearson correlation between hand size and sex. These are results, just move to the results section.

I understand that this kind of studies are difficult to develop and explain and this situation is reflected in the discussion. The results are clearly explained within the article; however, the discussion is the opportunity to contrast the result with other studies. Try to include some references showing a contrast with similar studies.

Finally, I can see the hypothesis was accepted. I don´t know if the study was designed/planned to say what was the better device.

Author Response

Thank you for the thorough and thoughtful review. We have carefully considered all your comments and have acted upon them, with changes in the manuscript that have contributed to an improved manuscript. Thank you!

The article is very interesting and complete; however, it needs to be restructured.

Thank you. Based on your recommendations, we have improved the background, improved the description of the methods, made the presentation of results clearer and have developed the conclusions with explicit support from the results.

- In the materials and methods section, I suggest using a better structure: study design, variables, participants, test tasks, and data analysis. In lines 74-81, the authors mention some details about statistical procedures, but it is not clear what non-parametric test they used and the level of significance. In the results Kendall´s W is mentioned but not in the methods.

Thank you for this constructive feedback. We have  restructured the Materials and Methods section adding the subsections as you suggested and have also clarified the statistical tests deployed in the analysis. 

- Try not to mix results in the materials and methods section. For example, Table 2 shows the results of hand anthropometry, and line 79 shows the result of the Pearson correlation between hand size and sex. These are results, just move to the results section.

Done! We have transferred hand anthropometric data and correlation analysis to the Results section.

- I understand that this kind of studies are difficult to develop and explain and this situation is reflected in the discussion. The results are clearly explained within the article; however, the discussion is the opportunity to contrast the result with other studies. Try to include some references showing a contrast with similar studies.

Any other study that has tested different sizes of devices with similar inclined geometry is unknown, that is, with angular amplitude between 50 and 60 degrees. Nevertheless, the results obtained here allow to establish a high level of qualitative similarity with the results obtained in references [6] and [8]. Both suggest that smaller PC mice compared to the users' hand size allows reaching higher levels of usability.

- Finally, I can see the hypothesis was accepted. I don´t know if the study was designed/planned to say what was the better device.

Yes, the study was designed with the goal of identifying the effect of size on usability for slanted devices. Considering the results obtained in terms of subjectively evaluated usability dimensions, and the results obtained from calculated efficiency, the study suggests that for the same inclination it is preferable to choose a smaller size. this conclusion is nuanced by the proportionality rule, as it is the medium-sized slanted device that obtains the best results in the evaluations related to the larger hand size group and the small-sized slanted device that is better for the smaller hand size participants.

Reviewer 2 Report

The paper deals with the interesting topic of usability issues of a work device (i.e. slanted computer mice). This is well written and organised, but there are some drawbacks to be improved, mainly regarding the possibility to better describe some aspects and provide further details to the readers. I suggest major revisions. Please, see the PDF attached.

Author Response

Reviewer 2

The paper deals with the interesting topic of usability issues of a work device (i.e. slanted computer mice). This is well written and organised, but there are some drawbacks to be improved, mainly regarding the possibility to better describe some aspects and provide further details to the readers. I suggest major revisions. Please, see the PDF attached.

Thank you for the thorough and thoughtful review you kindly carried out. We have carefully considered all your comments and have acted upon them, with changes in the manuscript that have contributed to a much-improved manuscript. Thank you!

Introduction

The Introduction section should be enriched in order to provide other interesting details about the motivation and context of the research. For instance:

  • the authors quote "usability advantages for slanted devices": please, quote some examples; • give other useful references of the pivotal elements of the study: usability, slanted computer mice, ergonomics;
    • the size of the device is underlined as one of the main factors affecting the usability: is that the only parameter to focus on?.

For enriching this section, I suggest the reading of the following references [removed for conciseness]:

Additionally, are there any International standards specifically developed for usability issues?

Thank you for this very constructive feedback. We have made the introduction more complete adding both general and specific information. We have added some new references to an entirely new paragraph:

The conventional (horizontal) PC mouse leads the user to uphold an unnatural posture, forcing the forearm to pronate until the end of the joint rotation amplitude. The PC mouse is most likely the most widely used hand tool in the world; the conventional geometry appeared about 50 years ago and the discomfort verified during its use was considered normal, despite the forearm and wrist posture that had to be adopted contrasting with the handwriting posture, learned at school from an early age. A slanted PC mouse, when properly sized, leads to a more neutral posture of the forearm, between pronation and supination. “Good design enables equipment to be used with the joints in the middle of their range of movement” [A]. Moreover, concepts such as ease of use and satisfaction, in addition to effectiveness and efficiency, are considered an integral part of a broader concept called usability [B]. A slanted handheld pointing device, provided it is properly dimensioned, will promote a more neutral posture of the forearm and wrist, with less effort required to operate and less discomfort being felt in its use compared to the conventional PC mouse, providing a usability advantage. In a broader approach, and following the principles of ergonomics, this study seeks to contribute to improving working conditions with regard to human well-being, safety and health [C].

[A] - Bridger R. Introduction to ergonomics. CRC Press; 2008.

[B] - International Organization for Standardization. ISO 9241: Ergonomics of human-system interaction - Part 11: Usability: Definitions and concepts. 2018

[C] - International Organization for Standardization. ISO 6385: Ergonomics principles in the design of work systems. 2016

Usability: Ref. [B] and [E] has been included – see further down.

Slanted PC mice: These are referred to in the Materials & Methods section, and Refs. [13, 14 and 15]. Consider the need to construct a sentence, to put in the introduction, that contains any of these references.

Ergonomics: Ref. [C] has been included

The degree of inclination is also an important usability factor in this type of handheld devices, Refs. [2], [7], [13] and [D].

[D] - Hedge A, Feathers D, Rollings K. Ergonomic comparison of slanted and vertical computer mouse designs. InProceedings of the Human Factors and Ergonomics Society Annual Meeting 2010 Sep (Vol. 54, No. 6, pp. 561-565). Sage CA: Los Angeles, CA: SAGE Publications.

As for applicable regulations regarding Usability, the ISO 9241-11 (2018) standard: Ergonomics of human-system interaction [E] presents concepts and definitions in this context. This standard defines usability as “the extent to which a system, product or service can be used by specified users to achieve specified goals with effectiveness, efficiency and satisfaction in a specified context of use”, provides annexes A and B that inform about how usability can be considered for different scopes of contexts of use and provide examples of usability measures.

On the other hand, the ISO 9241-400 (2007) standard: Ergonomics of human-system interaction [E] refers to the Principles and requirements for physical input devices. However, it is considered that it does not have a sufficiently structured process, which facilitates its application, to evaluate the usability of computer mice. Thus, with regard to assessing the usability of this type of devices, the authors developed specific tools to assess effectiveness, efficiency and satisfaction [2, 5 and 7].

[E] - International Organization for Standardization. ISO 9241: Ergonomics of human-system interaction – part 400: Principles and requirements for physical input devices. 2007

Materials and Methods 
Please provide other details about the participants (e.g. age, nationality). Table 1 is very interesting, but lacking of some pieces of information, such as dimensions and buttons.

Information on the subjects age and nationality has been provided in the subsection on participants (mean age 23.1 years (sd=6.5); Portuguese nationals and nationals from Portuguese speaking countries studying in Portugal)       Table 1 was enriched with the dimensions of the devices (similar arrangement of buttons is now referred in the text ).

Microsoft 1850 –                      100.0x58.1x38.2 (LxWxH in mm)

Moko S8 -                              104.9 x 72.9 x 59.9 (LxWxH in mm)

CSL E.VE –                             126.0 x 68.1 x 62.0 (LxWxH in mm)

Anker 98ANWVM –                120 x 62.8 x 74.8 (LxWxH in mm)

Buttons: Microsoft-3, Moko-6, CSL-6, Anker-5

...arrangement of buttons is now referred in the Table – subjects were instructed to only use the left and right main buttons as well as the middle button which were similar across all the devices, even if some had additional buttons that did not activate the graphical tests however

Results

Are there any support references for equations (1), (2), and (3)? I think that all the description of these equations should be moved in the section dedicated to materials and methods considered and carried out for this research. Could the authors report some results about the duration of the several tasks?

The equations spring from previous work of the lead authors and are now referenced appropriately [2,5 and 7].

This is added to the main text (section 2.): The tasks were designed with an approximate duration of one minute. However, the number of mistakes made by the participants influences the duration of the respective task, due to the requirement for its completion (number of targets).

Discussion

The Discussion section requires further work: the authors should better investigate the results of their work, compare them with what exists in the literature, the implications of the results, the strengths and limitations of the study, and possible future researches for addressing the current limitations of the study.

The existence of other results with these same devices and using these assessment instruments is unknown, and precludes establishing a comparison.

Manufacturers should produce at least two or more sizes of the same type of device (maintaining the angular tilt range).

Limitations already reported in the previously submitted version:

- Imbalance between the number of participants in the hand size categories for which an objective usability assessment was made;

- The weighted usability efficiency indicator, used to aggregate efficiency measures, was developed taking into account CAD tasks, therefore, its generalization to a broader set of office tasks is also limited.

Other limitations (now added):

- Lack of experience and training of the subjects with any slanted devices;

- The slanted devices that were tested do not have exactly the same shape and inclination;

Strong points:

- Results from a wide set of dimensions (measures) on efficiency and satisfaction (usability) supported by statistically significant tests.

- The subjects are habitual users of PC mouse (they are students in courses with a strong use of this type of manual devices).

Future studies (already reported in the previously submitted version of the manuscript):

- Refining efficiency weights for specific activities based on observational data;

- Conducting comparative assessments on a broader set of slanted devices;

- Studying muscle activation in the forearm, arm and shoulder in order to support the development of recommendations aimed at improving the working environment of users of these manual devices.

Future studies (added in the current version of the manuscript):

- Test different sizes of the same model (inclined at approximately 60 degrees);

- Predict the duration of the period of adaptation of the subjects with the devices under test;

- Increase the number of participants and the diversity in age and cultural origin.

Conclusions

This section should contain some quantitative results obtained in the study. Furthermore, the authors should highlight the audience of their work: for instance, which category of people (e.g. researchers, practitioners, designers of the device) might be interested in the results?

Salient quantitative results from the study include the high correlation factor (0.822) verified among subjective performance and satisfaction usability parameters. This is contrasted by smaller values of correlation factor when associating objective and subjective usability parameters (e.g. weighted efficiency versus effortlessness with r=0.418 and a p-value of 0,017). Mean values of weighted efficiency recorded in the study range from 68% to 75%, showing small differences across devices but which are inherently coherent with the preference rank orders.

The following audiences may be interested in the results of this work:

- Entities that develop and or that produce PC mice.

- Users who make intensive use of PC mice.

- Regulators and work environment authorities establishing recommendations to promote working health and extended social sustainability.

Round 2

Reviewer 2 Report

I appreciated the efforts carried out by the authors for the improvement of the paper. My final suggestions regarding the following aspects:

  • to homogenise the pieces of information provided in the Introduction section (e.g. the aim of the paper in rows 57-58 with rows 73-75, the usability notions in rows 61-72 with rows 83-89) and report the references in the style of the journal;
  • the current Table 5 is in reality Table 2;
  • please, review all the numbering of the tables;
  • please, review the numbering of the references;
  • to choose the most suitable position of the current Table 2 (Materials and Methods section or Result section).

Author Response

Thank you for the thorough feedback provided on the revision and for the additional comments which are very helpful and constructive.

I appreciated the efforts carried out by the authors for the improvement of the paper. My final suggestions regarding the following aspects:

to homogenise the pieces of information provided in the Introduction section (e.g. the aim of the paper in rows 57-58 with rows 73-75, the usability notions in rows 61-72 with rows 83-89) and report the references in the style of the journal;

Thank you for this pertinent remark. We have taken the aim quality to the sentence in lines 57-58 and made it more as a broad statement of alignment rather than a goal for the paper to remove any tension with the stated aims in lines 73-75.

We have compatibilized the usability notions in lines 83-89 with the ones in 61-72, by emphasizing the conceptual, high level theoretical nature of the former and the detailed, applied nature of the latter.

All references have been returned to the format in use by the journal.

the current Table 5 is in reality Table 2 ;

Thank you, we have changed the caption to Table 2, as well as the mention in the text (Table 2).

please, review all the numbering of the tables;

Table 2 is now disambiguated, and there is now a correct order in the numbering of the  tables and their captions as well as cross referencing in the text. 

please, review the numbering of the references;

Apologies, for there had been a problem in the submission of the first revision, in that we did a mistake and did not submit the latest version where the references had been streamlined and corrected, which is now the case for this second revision.

to choose the most suitable position of the current Table 2 (Materials and Methods section or Result section).

We have taken the reference to anthropometric data away from methods and it is now only in results.